# The activity of the aryl hydrocarbon receptor in T cells tunes the gut microenvironment to sustain autoimmunity and neuroinflammation

**Andrea R. Merchak**[1,2,3], **Hannah J. Cahill**[1], **Lucille C. Brown**[1], **Ryan M. Brown**[1,2,3], **Courtney Rivet-Noor**[1,2,3], **Rebecca M. Beiter**[1,2,3], **Erica R. Slogar**[1], **Deniz G. Olgun**[1], **Alban Gaultier** [1,3]*

1 Department of Neuroscience, University of Virginia, Charlottesville, Virginia, United States of America,
2 Neuroscience Graduate Program, University of Virginia, Charlottesville Virginia, United States of America,
3 Center for Brain Immunology and Glia, University of Virginia, Charlottesville, Virginia, United States of America

* ag7h@virginia.edu

**Data Availability Statement:** The datasets supporting the conclusions of this article are available in the NCBI Gene Expression Omnibus

## Abstract

Multiple sclerosis (MS) is a T cell-driven autoimmune disease that attacks the myelin of the central nervous system (CNS) and currently has no cure. MS etiology is linked to both the gut flora and external environmental factors but this connection is not well understood. One immune system regulator responsive to nonpathogenic external stimuli is the aryl hydrocarbon receptor (AHR). The AHR, which binds diverse molecules present in the environment in barrier tissues, is a therapeutic target for MS. However, AHR's precise function in T lymphocytes, the orchestrators of MS, has not been described. Here, we show that in a mouse model of MS, T cell-specific *Ahr* knockout leads to recovery driven by a decrease in T cell fitness. At the mechanistic level, we demonstrate that the absence of AHR changes the gut microenvironment composition to generate metabolites that impact T cell viability, such as bile salts and short chain fatty acids. Our study demonstrates a newly emerging role for AHR in mediating the interdependence between T lymphocytes and the microbiota, while simultaneously identifying new potential molecular targets for the treatment of MS and other autoimmune diseases.

## Introduction

Multiple sclerosis (MS) is a chronic inflammatory demyelinating disease that affects 2.3 million people worldwide. MS etiology and pathology are linked to both genetic and environmental factors [1]. A primary environmental factor impacting disease pathology lies with the gut microbiota [2–5]. Indeed, the connection between the microbiota and the immune system in autoimmune disorders such as MS is well established [6]. The gut flora, composed of bacteria, fungi, and viruses, aids food digestion and maintains pathogen control, but recent studies have also highlighted the integral role of the microbiome in modulation of the homeostatic immune

repository accession number GSE200440 and in additional supplementary files. https://www.ncbi.nlm.nih.gov/geo/query/acc.cgi?acc=GSE200440.

**Funding:** This work received funding from the National Institutes of Health (R33 MH108156 to A. G., T32 NS115657 to A.R.M., T32 GM008136 to R. M.B.), from the UVA Wagner fellowship (R.M.B.), from the Owens Family Foundation (A.G.) from the UVA Trans University Microbiome Initiative pilot grant (A.G. and A.R.M.) and from the UVA Presidential Fellowship in Neuroscience (C.R.N.). The funders had no role in study design, data collection and analysis, decision to publish, or preparation of the manuscript.

**Competing interests:** The authors have declared that no competing interests exist.

**Abbreviations:** AHR, aryl hydrocarbon receptor; ASV, amplicon sequence variant; BA, bile acid; CNS, central nervous system; EAE, experimental autoimmune encephalomyelitis; FMT, fecal material transfer; HE, hematoxylin and eosin; MS, multiple sclerosis; NK, natural killer; SCFA, short-chain fatty acid; sPLS-DA, sparse partial least squares discriminate analysis.

state both locally and systemically [7]. Gut dysbiosis is a hallmark of disease in both patients with MS and in experimental animal models of MS [2,5,8–10]. The dysbiotic microbiome can cause earlier onset and increase disease severity in mice [10]. Furthermore, germ-free mice lacking a microbiome are resistant to experimental autoimmune encephalomyelitis (EAE), an animal model of MS [11]. These findings spurred the examination of various bacterial supplements in mouse models of MS, and numerous potential candidates have been identified [6,12]. However, the mechanism(s) of action is not well understood. The importance of the microbiome in MS is well described, but a better understanding of the cross-talk between the immune system and the microbiome is required to translate these findings to the clinic. Further, discoveries in the context of MS provide a foundation for understanding environmental factors impacting many other autoimmune disorders.

The aryl hydrocarbon receptor (AHR) is a prime candidate for understanding the microbiome-immune interface. The AHR is a cytoplasmic receptor that, when activated, traffics to the nucleus to execute downstream transcriptional programing [13]. While canonical immune sensors have been primarily characterized based on their response to pathogenic material, the AHR is a homeostatic regulator that is activated by a variety of nonpathogenic exogenous ligands present in barrier tissues. These ligands include indoles [14], kynurenines [15], and other small molecules [16]. Downstream immune effects are dependent on the cell type and ligand. The AHR has also been directly tied to MS. AHR activation is the suspected mechanism by which the novel MS therapeutic, laquinimod, acts [17,18]. It is thought that AHR activation by laquinimod in antigen presenting cells reduces the ratio between inflammatory T cells and regulatory T cells (Treg). Microbiome changes have been identified in AHR null mice and in mice treated with a potent AHR agonist 2,3,7,8-tetrachlorodibenzo-p-dioxin (TCDD), further supporting the connection between AHR, the microbiome, and MS [19–23]. While AHR activity in response to microbiome derived metabolites is well described, a gap of knowledge remains about the impact of AHR expression on the microbiota composition and function.

Here, we explore the role of AHR activity in CD4+ T cells using EAE. We have demonstrated that deleting *Ahr* from CD4+ T cells increases recovery in EAE in a microbiome-dependent manner. This improved recovery was the result of increased T cell apoptosis after activation in the central nervous system (CNS). While AHR deficiency did not grossly alter the composition of the microbiome, it significantly impacted the production of microbiome-mediated metabolites. In particular, AHR deficiency in T cells led to an increase in bile acids (BAs) and a subset of short-chain fatty acids (SCFAs) that ultimately impacted T cell viability. Our study aimed to understand how a microbiota response element can act in the inverse in the context of autoimmunity. This is the first demonstration, to our knowledge, of a role for AHR in T lymphocytes as a regulator of the microbiome activity that ultimately influences the outcome of CNS autoimmunity. Our discovery builds a foundation for further work that could lead to a microbiome centric approach to dampen the overactive immune system in MS and related autoimmune disorders.

## Results

### Separately housed CD4+ specific *Ahr* knockout mice recover from active EAE

Given the emerging role of the microbiome and the well-accepted contribution of T cells to MS pathology, AHR is ideally positioned to influence CNS autoimmunity. Here, we explored the role of AHR in CD4+ T cells in EAE. For this study, we used the previously validated $Cd4^{cre}Ahr^{fl/fl}$ and littermate control $Ahr^{fl/fl}$ mouse strains [24]. Cohoused adult littermate mice were immunized with myelin oligodendrocyte glycoprotein (MOG)$^{35-55}$ to induce active EAE.

The cohoused mice showed no difference in EAE clinical score when *Ahr* expression was removed from CD4+ cells (**Figs 1A** and **S1A**, females; **S1B and S1C Fig,** males). Further, we observed no difference in myelin content in the spinal cord, as revealed by Luxol fast blue staining at the chronic phase of disease (**Fig 1B and 1C**). To assess the role of AHR-driven microbiome changes in EAE, we separated mice by genotype at weaning (3 weeks) and induced EAE between 8 and 16 weeks of age. In this condition, while the time of onset and the peak clinical scores of EAE were not impacted, mice lacking *Ahr* expression in T cells presented with a significant recovery when compared to wild-type animals (**Figs 1D** and **S1D**, females). Furthermore, the change in recovery does not appear to be driven by sex as it was conserved in both sexes (**S1E and S1F Fig**, males). Supporting the clinical scores, the demyelination of the spinal cords was not different between the 2 groups at the peak of disease (day 16); however, we noted a significant increase in myelin staining at the chronic phase (day 31) in the *Cd4^{cre}Ahr^{fl/fl}* animals when compared to the control (**Fig 1E–1G**). Together, these data indicate that the lack of *Ahr* expression in CD4+ cells promotes EAE recovery in a microbiome-dependent manner that can be reversed by cohousing.

## T cell differentiation and activity are unchanged in *Cd4^{cre}Ahr^{fl/fl}* mice

CD4+ T cells are responsible for orchestrating EAE pathology, and defects in these cells have been shown to impact disease outcomes [25]. Additionally, AHR agonists can modulate T cell skewing in a ligand dependent manner in vivo [26]. We aimed to explore the role of AHR in T cell differentiation and cytokines production. First, we examined T cell differentiation and the production of cytokines in cohoused *Cd4^{cre}Ahr^{fl/fl}* mice. We found that the deletion of AHR in T cells did not impact their capacity for in vitro differentiation as determined by expression of lineage transcription factors by flow cytometry (**Fig 2A–2E**) or production of the signature cytokine for each lineage by ELISA (**Fig 2F–2I**). This was mirrored in separately housed animals (**S2A–S2G Fig**). Furthermore, there were no differences in the immune cell composition between the knockout and control animals at baseline in gut associated lymphoid tissue (**Fig 2J–2M**). We next measured several cytokines important to EAE induction in the spinal cords of mice at the peak of disease. We expected to see significant reduction in the pathological cytokines in the separately housed *Cd4^{cre}Ahr^{fl/fl}* mice that ultimately recover, but surprisingly we found minimal differences (**Fig 2N–2S**). The only change was a small reduction in TNFα that may warrant further investigation. These findings suggest that CD4 cell intrinsic AHR activity is not necessary for normal T cell development and cytokine expression supporting published work indicating that AHR activation of the antigen presenting cell may be more important to T cell subset generation in vivo [27–29]. Further, these data reinforce the hypothesis that the T cell-intrinsic reduction of AHR activity in *Cd4^{cre}Ahr^{fl/fl}* mice is not responsible for EAE recovery described in separately housed mice.

## *Cd4^{cre}Ahr^{fl/fl}* mice have fewer T cells in the spinal cord during EAE

To determine whether EAE recovery in *Cd4^{cre}Ahr^{fl/fl}* was linked to a difference in immune cell number during pathology, we quantified CD45+ immune cells in the spinal cord tissue using immunohistochemistry at the peak and chronic stages of EAE progression (**Fig 3A and 3B**). While the quantity of CD45+ cells was equivalent at the peak of the disease, we observed a decrease in the number of CD45+ cells in the *Cd4^{cre}Ahr^{fl/fl}* spinal cord compared to controls at the chronic phase (**Fig 3B**). We next aimed to determine whether this reduction was a result of changes in the number of infiltrating T cells or macrophages. To accomplish this, we quantified expression of CD3, a pan-T cell marker, by immunohistochemistry. We saw no difference in the CD3+ cells in the spinal cord at the peak of disease (**Fig 3C**); however, at the chronic

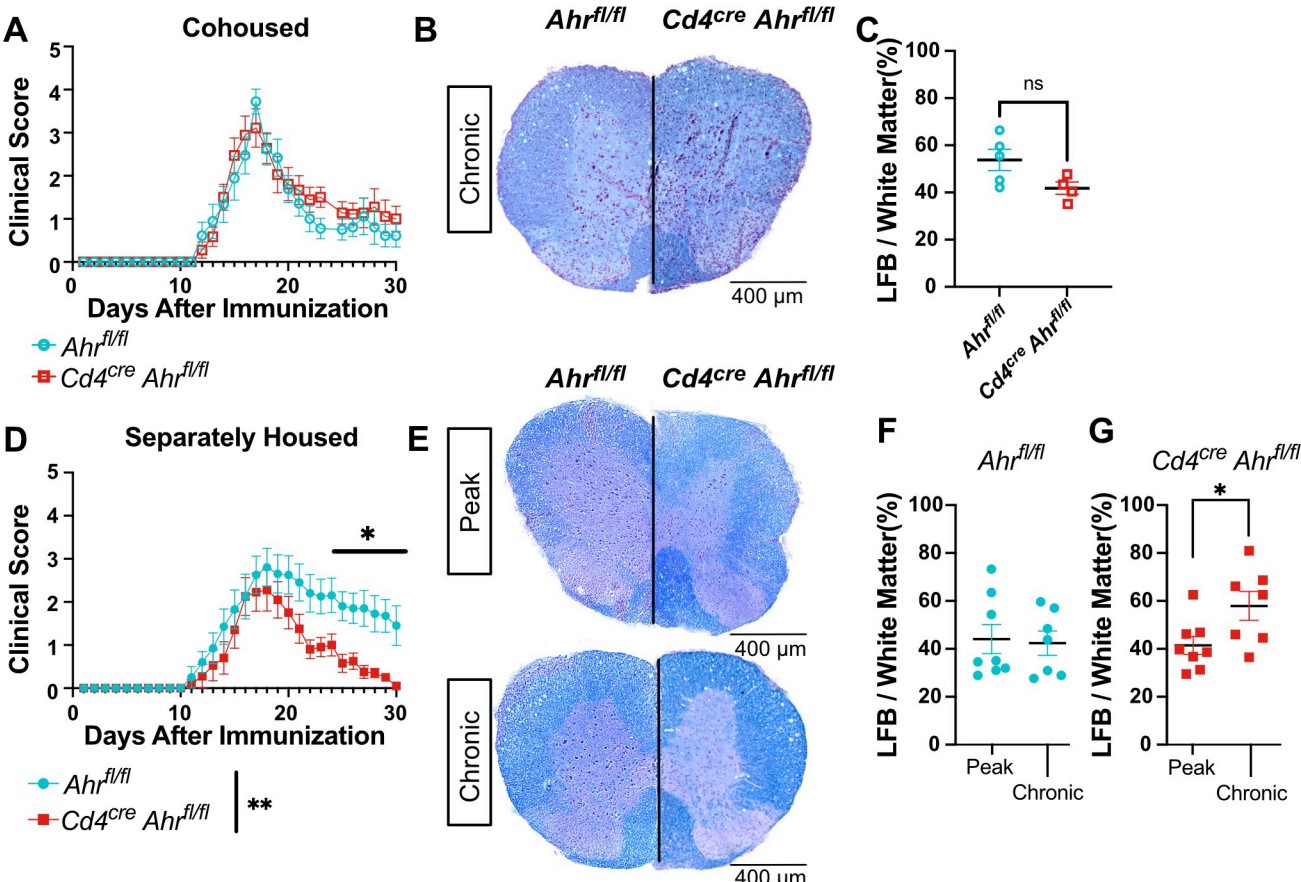

**Fig 1. Separately housed *Cd4^cre^Ahr^fl/fl^* recover from EAE with increased myelin staining at chronic phase.** (**A**) Clinical score of *Cd4^cre^Ahr^fl/fl^* and *Ahr^fl/fl^* mice cohoused (females; representative plot includes *n* = 9 mice/group; total replicates of *N* = 2 experiments). Spinal cords sections of *Cd4^cre^Ahr^fl/fl^* and *Ahr^fl/fl^* were stained with Luxol fast blue and hematoxylin/eosin stain at day 31 post EAE induction (**B**) Representative images and (**C**) quantification of myelin stain. Images from 4 equally spaced spinal cord levels were averaged for each mouse. (*n* = 5 mice/group; unpaired *t* test *p* = 0.0706) (**D**) Clinical score of *Cd4^cre^Ahr^fl/fl^* and *Ahr^fl/fl^* littermate controls separated at weaning (3 weeks of age). (Females; representative plot includes *n* = 8–9 mice/group; total replicates of *N* = 2 experiments; Mann–Whitney U test on total scores reported in legend [*p* = 0.0096] and on single days reported on plot) (**E**) Luxol fast blue with hematoxylin/eosin stain at the peak stage of EAE (day 16) and at chronic phase (day 31). (**F**) Quantification of myelin stain by Luxol fast blue alone in *Ahr^fl/fl^* mice and (**G**) *Cd4^cre^Ahr^fl/fl^* mice. (*n* = 7–8 mice/group; *N* = 2 experiments; unpaired *t* tests [*p* = 0.8343, 0.0322]) Scale bars represent 400 μm. Error bars represent standard error from the mean. Raw data can be found in Supporting information (S1 Data). EAE, experimental autoimmune encephalomyelitis.

phase, significantly lower CD3+ T cell coverage was observed in the *Cd4^cre^Ahr^fl/fl^* mice compared to the separately housed littermate controls (**Fig 3D**). Meanwhile, monocytes and macrophages, the other primary effector cells in EAE, showed no differences at either peak or chronic phase as shown by CD11b expression (**Fig 3E and 3F**). We confirmed this outcome using flow cytometry on Percoll-isolated immune cells from the spinal cord. At the peak of disease, we again observed no differences in infiltrating T cells (CD4+ or CD8+), T cell subsets (RORgt+, Tbet+ GATA3+, or FoxP3+), or macrophages (CD11b+) (**Fig 3G and 3H**). At the chronic phase of disease, we found that the loss of T cells was limited to the CD4+ compartment and not the CD8+ compartment. Of the classic CD4 helper cell subtypes, all trended down in *Cd4^cre^Ahr^fl/fl^* mice with no one subtype constituting a significant decrease, indicating a pan-CD4+ phenotype, not associated with a specific subtype of CD4+ T cells **Fig 3I**. Taken together, our data suggest that in separately housed mice, EAE recovery observed in *Cd4^cre^Ahr^fl/fl^* mice is not due to a difference in cytokine expression or T cell differentiation.

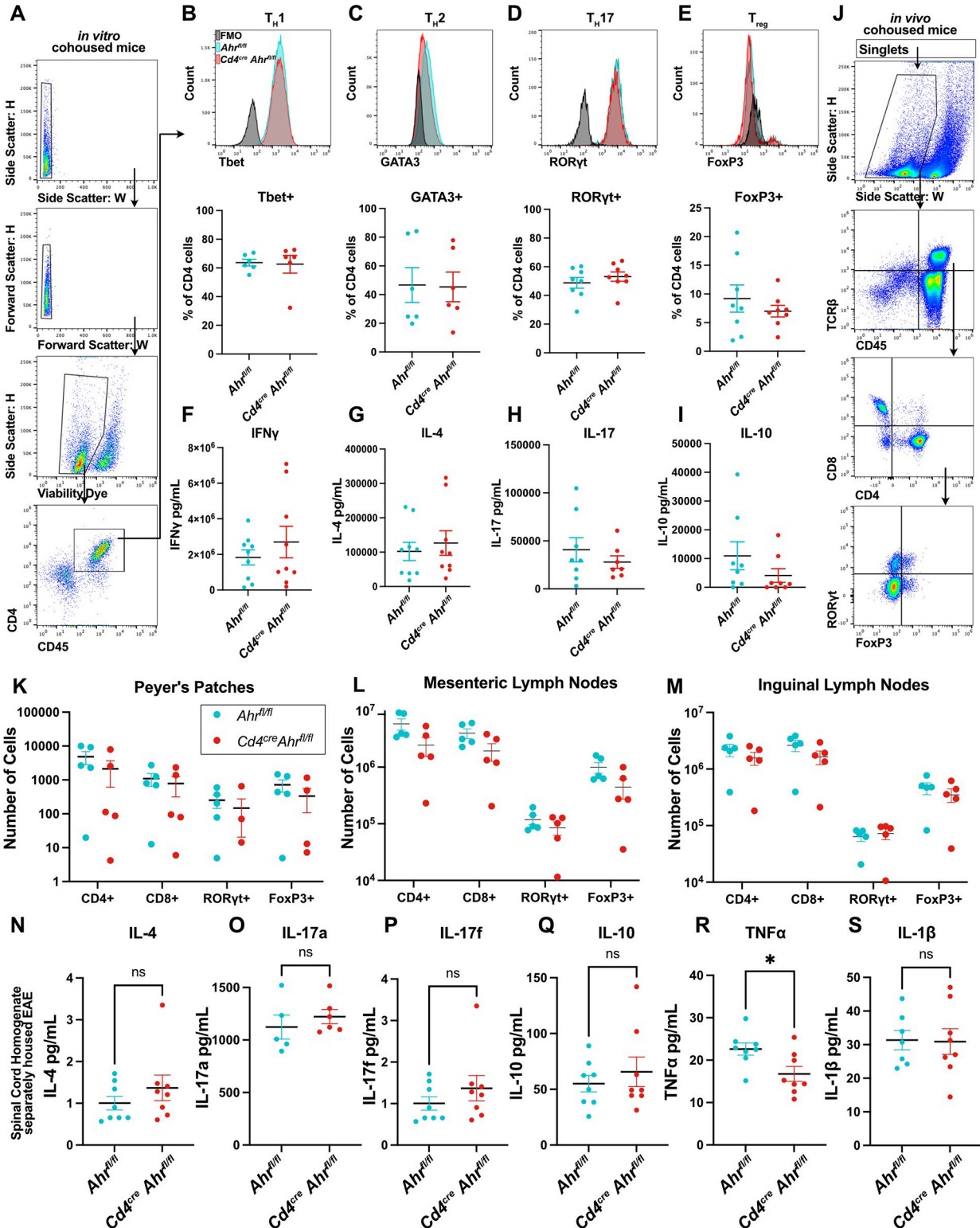

**Fig 2. *Ahr* knockout in CD4+ cells does not influence T cell differentiation.** (**A**) Naïve CD4+ T cells were isolated from cohoused *Cd4^{cre}Ahr^{fl/fl}* and littermate control animals and treated to promote differentiation. Flow cytometry gating strategy to measure percent of cells expressing the transcription factors specific to each cell type is shown. (**B**) Representative histogram and quantification of Tbet+ cells after in vitro differentiation of naïve CD4+ cells to $T_H1$ cell type, ($n = 6$ biological replicates/group; $N = 2$ experiments; unpaired $t$ test [$p = 0.8715$]) (**C**) GATA3 staining of $T_H2$ differentiated cells, ($n = 6$ biological replicates/group; $N = 2$ experiments; unpaired $t$ test [$p = 0.9356$]), (**D**) RORγt staining of $T_H17$ cells

($n$ = 8 biological replicates/group; $N$ = 3 experiments; unpaired $t$ test [$p$ = 0.3989]) (E), and FoxP3 staining for $T_{regs}$ ($n$ = 8 biological replicates/group; $N$ = 3 experiments; unpaired $t$ test [$p$ = 0.4078]). (F) ELISA analysis of culture supernatant after 24 h of anti-CD3 stimulation in $T_H1$ cells ($n$ = 9 biological replicates/group; $N$ = 3 experiments; unpaired $t$ test [$p$ = 0.3924]) (G) $T_H2$ cells ($n$ = 8 biological replicates/group; $N$ = 3 experiments; unpaired $t$ test [$p$ = 0.5917]) (H) $T_H17$ cells ($n$ = 7–8 biological replicates/group; $N$ = 3 experiments; unpaired $t$ test [$p$ = 0.4013]) (I) or Treg cells. ($n$ = 8 biological replicates/group; $N$ = 3 experiments; unpaired $t$ test [$p$ = 0.2278]) (J) Gating strategy for (K) flow cytometry of single-cell suspensions of Peyer's patches from cohoused $Cd4^{cre}Ahr^{fl/fl}$ and $Ahr^{fl/fl}$ mice. (Multiple $t$ tests with Welch correction [$p$ = 0.070748, 0.081520, 0.325104, 0.080798]) (L) Analysis of cell compositions from mesenteric lymph nodes (multiple $t$ tests with Welch correction [$p$ = 0.127038, 0.179526, 0.444670, 0.118472]) (M) and the inguinal lymph nodes from cohoused $Cd4^{cre}Ahr^{fl/fl}$ and $Ahr^{fl/fl}$ mice (multiple $t$ tests with Welch correction [$p$ = 0.381964, 0.224164, 0.664161, 0.465348]). (N–S) Luminex assay on whole spinal cord homogenate of separately housed $Cd4^{cre}Ahr^{fl/fl}$ and $Ahr^{fl/fl}$ mice at the peak of disease (day 16 after immunization; $n$ = 8 mice/group; $N$ = 2 experiments; unpaired $t$ test), (N) [$p$ = 0.3087], (O) [$p$ = 0.4580], (P) [$p$ = 0.8313], (Q) [$p$ = 0.4941], (R) [$p$ = 0.0215], (S) [$p$ = 0.9333]. Error bars represent standard error from the mean. Raw data can be found in Supporting information (S1 Data).

Instead, recovery correlates with an overall reduction in CD4+ T cell numbers at the chronic phase of disease and suggests an impact for AHR and the microbiome on CD4+ T cell fitness.

## Apoptosis of *Ahr*-deficient CD4 T cells is dependent on gut-derived small molecules

We next investigated the mechanism behind the reduced T cell numbers in the spinal cords of $Cd4^{cre}Ahr^{fl/fl}$ mice during chronic EAE. We performed a gene expression analysis on whole spinal cord tissues isolated from two $Cd4^{cre}Ahr^{fl/fl}$ and two $Ahr^{fl/fl}$ animals at the peak of EAE using a qPCR array composed of 384 genes associated with neuroinflammation. We found that the expression of many genes involved in apoptosis were modulated in the spinal cord tissue of $Cd4^{cre}Ahr^{fl/fl}$ mice; in particular, we noted increased expression of *Fas*, *GrB*, *Casp3*, and *Casp9* and decreased expression of the antiapoptotic transcript of *Xiap*, indicating an increase in apoptosis in CD4-specific *Ahr*-deficient mice (S3A and S3B Fig). To test if T cells lacking AHR were undergoing apoptosis in EAE, we stained spinal cords sections for CD3 expression by immunofluorescence and TUNEL assay (Fig 4A). While we could detect some apoptotic T cells in the spinal cords of control mice subjected to EAE, CD3+TUNEL+ cell numbers were significantly increased in the spinal cords of *Ahr*-deficient animals (closed triangles; Fig 4B). We confirmed this observation by quantifying cellular death of isolated immune cells from the spinal cord at the peak of disease using flow cytometry. We observed a significant and specific increase in the percentage of dead CD4+ T cells without change in the percentage of dead CD8+ and or total TCRB+ cells between genotypes (Fig 4C). To further understand the mechanisms behind these observations, we focused examination on the primary pathogenic subset of CD4+ cells in EAE, the type 17 T helper cell ($T_H17$) [30]. To obtain $T_H17$ cells, we isolated naïve CD4+ cells from the lymph nodes and spleen, then cultured the cells in $T_H17$ differentiation conditions. Similar to the in vivo observations, $Cd4^{cre}Ahr^{fl/fl}$ $T_H17$ cells have a significant increase in apoptosis after mild in vitro stimulation by anti-CD3 as measured by Annexin V staining compared to littermate controls (Fig 4D). Increased $T_H17$ cell death was only observed in cells isolated from $Cd4^{cre}Ahr^{fl/fl}$ animals separated at weaning, as T cells isolated from cohoused $Cd4^{cre}Ahr^{fl/fl}$ mice did not display elevated apoptosis (Fig 4E). Taken together, these results support the hypothesis that gut microenvironment may be implicated in the *Ahr*-dependent T cell fitness phenotype. To directly test this hypothesis, we isolated metabolites (<3 kDa) from cecums of separately housed $Cd4^{cre}Ahr^{fl/fl}$ and $Ahr^{fl/fl}$ mice, and added them to in vitro skewed $T_H17$ cells prepared from C57BL6/J mice during the 24-h stimulation period. We next analyzed apoptosis by flow cytometry. $T_H17$ cells exposed to metabolites derived from the gut of $Cd4^{cre}Ahr^{fl/fl}$ mice had elevated apoptosis compared to cells exposed to metabolites from $Ahr^{fl/fl}$ microbiome (Fig 4F). Collectively, these data demonstrate that exposure to the microbial metabolites from $Cd4^{cre}Ahr^{fl/fl}$ mice is sufficient to induce early T cell apoptosis which likely contributes to recovery in EAE.

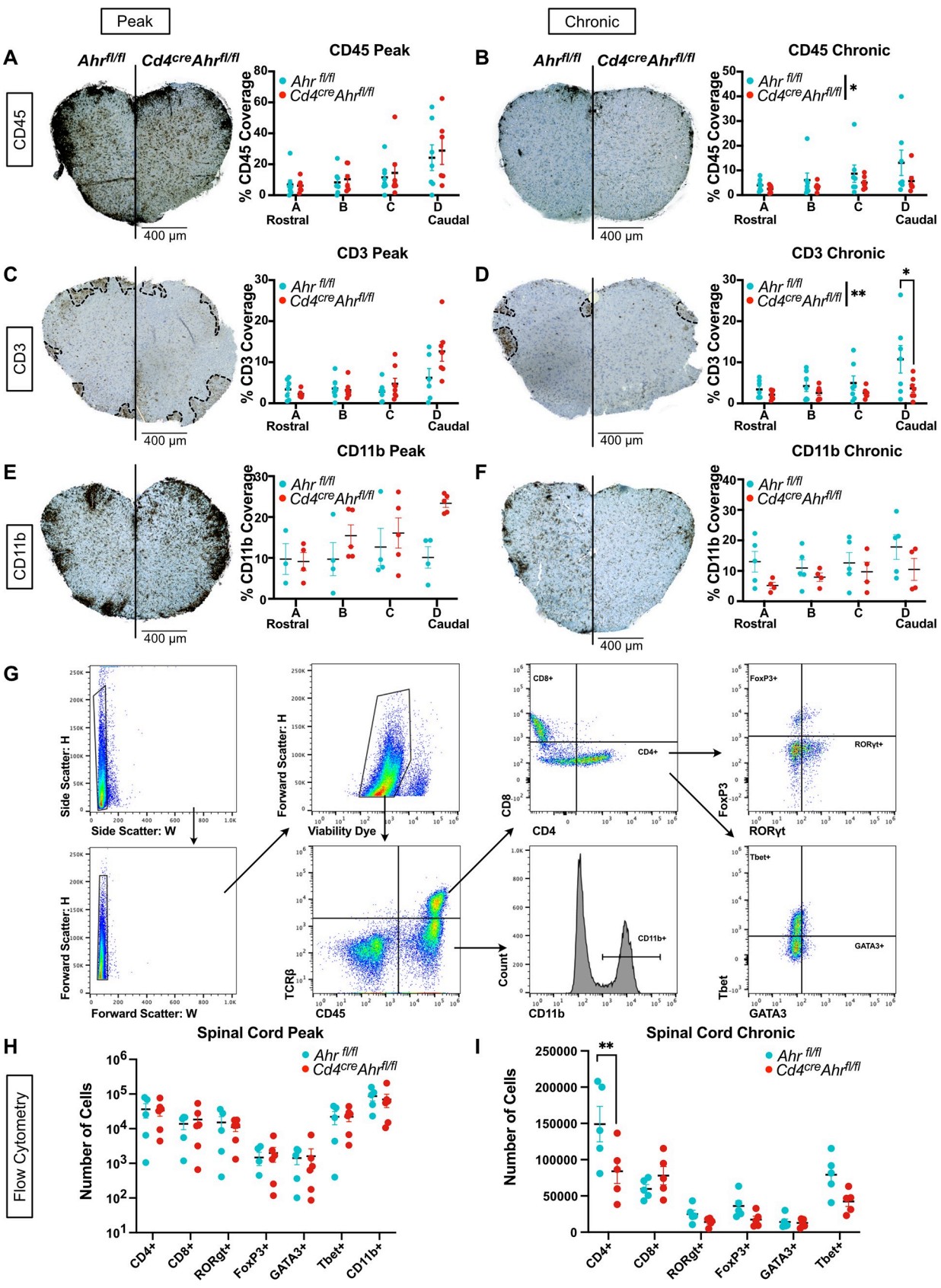

**Fig 3. Reduced CD3+ cells in the spinal cord of *Cd4^cre^Ahr^fl/fl^* mice correlate with clinical signs.** (**A**) Spinal cords of separately housed *Cd4^cre^Ahr^fl/fl^* and *Ahr^fl/fl^* littermate controls were stained for total immune cells (CD45+) by immunohistochemistry at 4 levels at the peak of disease (day 16 after immunization) (Each dot represents a mouse *n* = 7 mice/group; *N* = 2 experiments; two-way ANOVA [*p* = 0.5336]) (**B**) or at the chronic phase of disease (day 31 after immunization). (Two-way ANOVA [*p* = 0.0419]; Sidak's multiple comparisons test [*p* = 0.9896, 0.9209, 0.7898, 0.1753]). (**C**) Spinal cords of separately housed *Cd4^cre^Ahr^fl/fl^* and *Ahr^fl/fl^* littermate controls were stained for T cells (CD3+) by immunohistochemistry at 4 levels at the peak of disease (day 16 after immunization) (*n* = 7 mice/group; *N* = 2 experiments, two-way ANOVA [*p* = 0.0830]) (**D**) or at chronic phase of disease (day 31 after immunization). (*n* = 7 mice/group; *N* = 2 experiments, two-way ANOVA [*p* = 0.0054]; Sidak's multiple comparisons test [*p* = 0.9578, 0.9029, 0.7095, 0.0070]). (**E**) Spinal cords of separately housed *Cd4^cre^Ahr^fl/fl^* and *Ahr^fl/fl^* littermate controls were stained for macrophages (CD11b+) by immunohistochemistry at 4 levels at the peak of disease (day 16 after immunization) (*n* = 4–5 mice/group; *N* = 2 experiments; two-way ANOVA [*p* = 0.1271]) (**F**) or at chronic phase of disease (day 31 after immunization). (*n* = 5 mice/group; *N* = 2 experiments, two-way ANOVA [*p* = 0.2269]). (**G**) Gating strategy for flow cytometry analysis of immune cells isolated from dissociated spinal cord tissue (**H**) at peak of disease (day 16 after immunization) (*n* = 5–6/group; multiple unpaired *t* tests with Benjamini and Yekutieli corrections [*p* values listed in the Supporting information (S1 Data)]), (**I**) or chronic phase of disease (day 31 after immunization) (*n* = 5 mice/group; multiple unpaired *t* tests with Benjamini and Yekutieli corrections [*p* values listed in the Supporting information (S1 Data)]). Scale bars represent 400 μm. Error bars represent standard error from the mean. Raw data can be found in Supporting information (S1 Data).

## Metabolomic differences in the gut microenvironment are evident in separately housed *Cd4^cre^Ahr^fl/fl^* mice

To identify the impact of CD4+ cell AHR loss on the gut microenvironment, we began by performing 16S sequencing of DNA isolated from fecal pellets and cecal contents of *Ahr^fl/fl^* or *Cd4^cre^Ahr^fl/fl^* mice. Neither cohoused nor separately housed animals saw differences in fecal or cecal microbial signatures (S4A–S4D Fig). Given the marked effect of the metabolites prepared from the cecums of *Cd4^cre^Ahr^fl/fl^* and *Ahr^fl/fl^* on T cell apoptosis (Fig 4F), we next conducted untargeted metabolomics by LC-MS on these preparations (<3 kDa). Two-dimensional clustering of the annotated products revealed distinct clustering of genotypes, indicating different chemical profiles (Fig 5A). This difference occurred despite there being no significant difference in the 16S microbial signatures of the cecal contents suggesting that AHR activity in T cells modify microbial metabolism without altering the microbe composition to a degree that can be captured by 16S sequencing (S4E and S4F Fig). Impact analysis of biological pathways identified primary bile acid biosynthesis as the most significantly up-regulated pathway with 6 primary (cholic, taurocholic) or secondary (glycocholic, sulfocholic, dehydrocholic, lawsonic) bile acids significantly increased in the cecal microenvironment of *Cd4^cre^Ahr^fl/fl^* mice (Fig 5B and 5C). Bile acids are dysregulated systemically in patients with MS and adding back primary bile acid is sufficient to reduce the severity of EAE [31]. Bile acid biosynthesis is known to be dependent on the gut microbiota composition [32,33]. Of the changed bile acids, taurocholic acid was the most highly up-regulated with over a 10-fold increase in *Cd4^cre^Ahr^fl/fl^* compared to control mice (Fig 5C). Increased bile acid accumulation in these mice may be due to the down-regulation of bile acid transporters and response elements in the small intestine of the *Cd4^cre^Ahr^fl/fl^* mice (Fig 5D). In addition to changes in bile acids, alanine, aspartate, and glutamate metabolism were also up-regulated, whereas purine metabolism and tryptophan metabolism were down-regulated in *Cd4^cre^Ahr^fl/fl^* mice. As SCFAs have been reported to impact the viability of inflammatory T cell and cannot be captured through untargeted metabolomics, we next performed targeted SCFA metabolomics [34,35]. Isovaleric acid was the only SCFA with a significantly higher abundance in the *Cd4^cre^Ahr^fl/fl^* cecums compared to controls (Fig 5E).

## Taurocholic acid induces T$_H$17 cell apoptosis and protects from EAE

The chemical composition of the cecal microenvironment between separately housed *Cd4^cre^Ahr^fl/fl^* and *Ahr^fl/fl^* animals is significantly changed. We next aimed to determine what functional differences arose from the chemical differences both in vitro and in vivo. We first

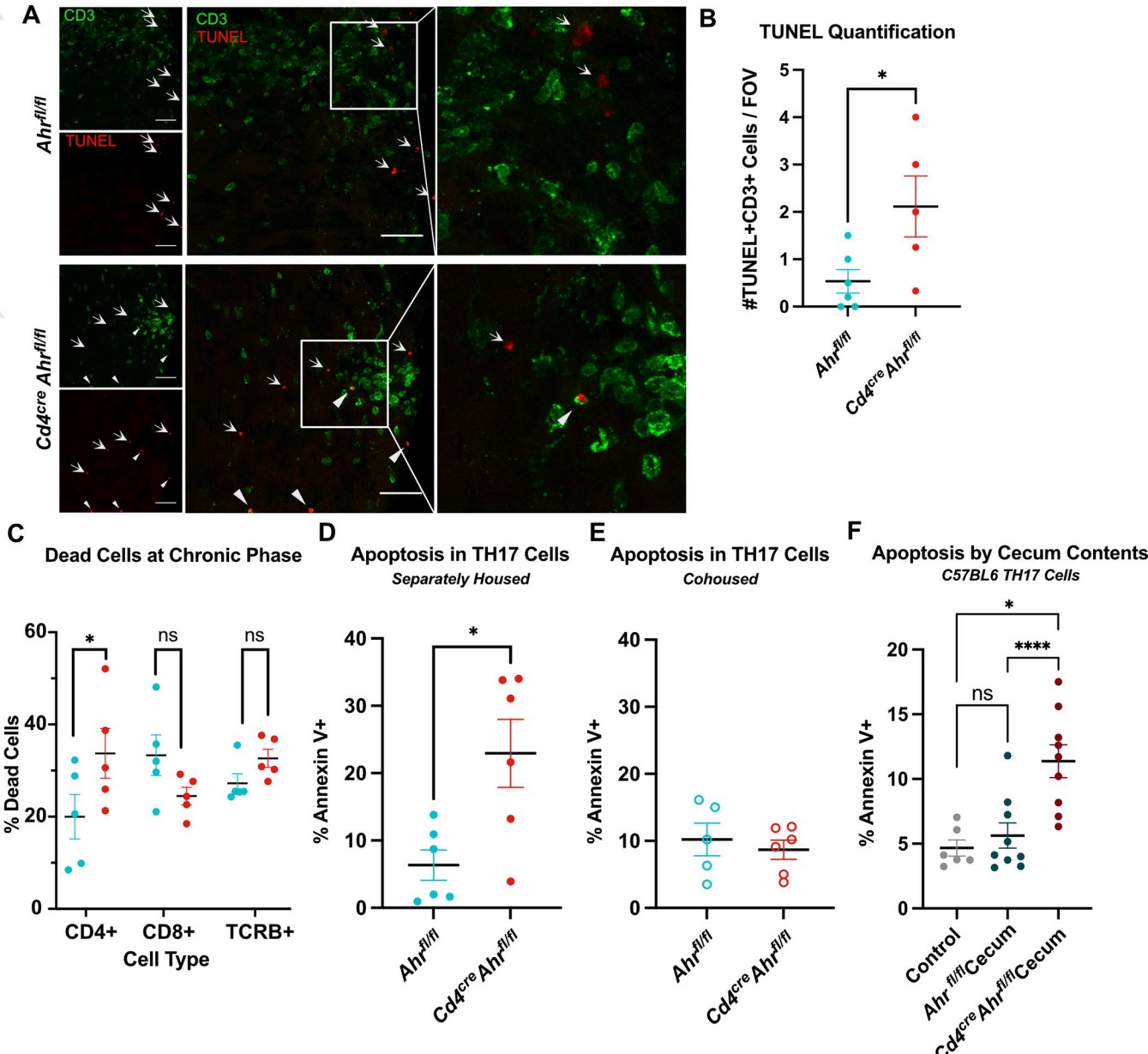

**Fig 4. T cell apoptosis is increased in animals who recover from EAE.** (**A**) Representative image of spinal cords stained with TUNEL assay and T cell marker (CD3) at the peak of disease. Closed triangles indicate TUNEL+CD3+ cells and arrows indicate TUNEL+CD3- cells. Scale bar = 25 μm. (**B**) Quantification of the number of TUNEL+ T cells (2–6 lesion sites averaged per animal; $n$ = 5–6 mice; $N$ = 2 EAE experiments; unpaired $t$ test [$p$ = 0.0357]). (**C**) T cells isolated from the spinal cord at the peak of disease stained with viability dye and measured by flow cytometry. Cells gated on singlets, CD45+, gate shown on x-axis, then dead cells ($n$ = 5 mice/group; two-way ANOVA [$p$ = 0.0192] with Sidak's multiple comparison tests [$p$ = 0.0465, 0.2862, 0.6800]). (**D**) T cells skewed to $T_H17$ in vitro from separately housed $Cd4^{cre}Ahr^{fl/fl}$ mice stained for apoptotic marker Annexin V 24 h after stimulation with anti-CD3 antibody. ($n$ = 6 mice/group; $N$ = 2 experiments; unpaired $t$ test [$p$ = 0.0133]). (**E**) $T_H17$ cells from cohoused $Cd4^{cre}Ahr^{fl/fl}$ and $Ahr^{fl/fl}$ mice show no differences in the number of Annexin V positive cells 24 h after stimulation with anti-CD3 antibody. ($n$ = 5–6 mice/group; $N$ = 2 experiments; unpaired $t$ test [$p$ = 0.5851]). (**F**) In vitro differentiated $T_H17$ cells from C57BL6/J mice were exposed to the <3 kDa fraction of cecal contents from $Ahr^{fl/fl}$ and $Cd4^{cre}Ahr^{fl/fl}$ mice for 24 h with anti-CD3 stimulation. ($n$ = 6–9 mice/group; $N$ = 3 experiments; mixed-effects analysis, with Geisser–Greenhouse correction [$p$ = 0.0013]; Tukey's post hoc analysis [Ctr vs. Fl $p$ = 0.7729; Ctr vs. Cre $p$ = 0.0333, Fl vs. Cre $p$ < 0.0001]). Error bars represent standard error from the mean. Raw data can be found in Supporting information (S1 Data). EAE, experimental autoimmune encephalomyelitis.

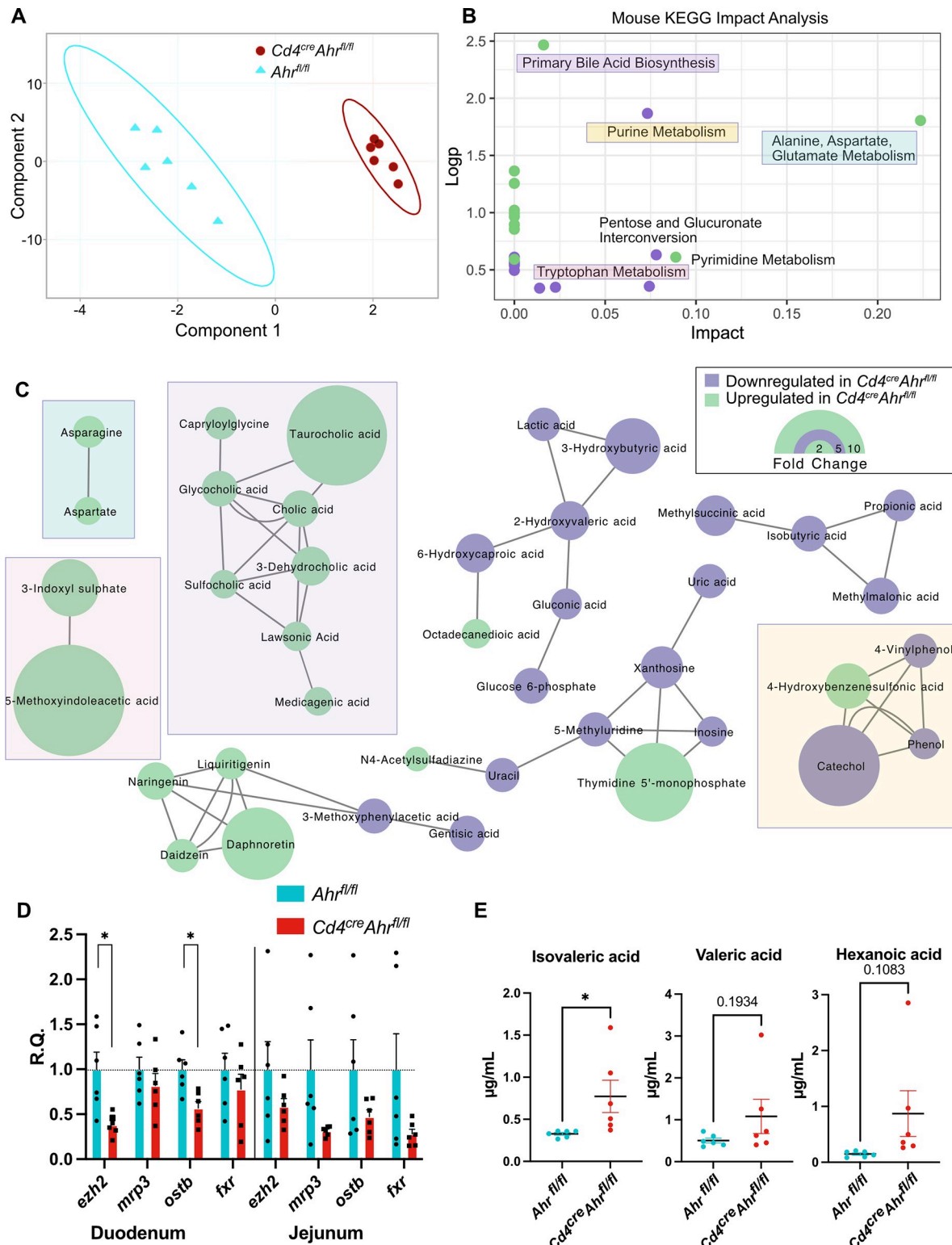

**Fig 5. Elevated bile salts and SCFA detected in the cecum of AHR-deficient mice.** (**A**) Partial least-squares plot of LC-MS untargeted metabolomics of the <3 kDa fraction of the cecal contents from *Cd4^cre^Ahr^fl/fl^* and *Ahr^fl/fl^* mice. (**B**) Metaboanalyst mouse KEGG pathways significantly changed by genotype. (**C**) MetaMapp network analysis visualizing the chemically or biologically related significantly changed molecules. (**D**) qPCR of bile acid transporters (*mrp3*, *ostb*) and bile acid response elements (*ezh2*, *fxr*) in bulk tissue of the duodenum and jejunum of separately housed animals illustrated as relative quantity (R.Q.) from *Ahr^fl/fl^* tissue. (*n* = 6 mice/group; *N* = 2 experiments;

multiple *t* tests with Welch correction [duodenum *p* = 0.0104, 0.3591, 0.0071, 0.3828; jejunum *p* = 0.2230, 0.0623, 0.1487, 0.1004]). (**E**) Quantitative targeted metabolomics for SCFAs show a significant increase in isovaleric acid and a trending increase in valeric and hexanoic acid in $Cd4^{cre}Ahr^{fl/f}$ mice. (*n* = 6 mice/group; *N* = 1 experiment; unpaired *t* tests [*p* = 0.0427, 0.1934, 0.1083]) Error bars represent standard error from the mean. Raw data can be found in Supporting information (S1, S2, and S3 Data files). AHR, aryl hydrocarbon receptor; SCFA, short-chain fatty acid.

examined the most highly increased bile acid in $Cd4^{cre}Ahr^{fl/fl}$ mice, taurocholic acid. When administered to C57BL6/J-derived $T_H17$ cells in vitro, taurocholic acid was sufficient to drive apoptosis similarly to cecal content isolates (**Fig 6A and 6B**). We also examined isovaleric acid, the SCFA, elevated in the absence of AHR. Similarly to taurocholic acid, isovaleric acid could induce apoptosis of in vitro generated $T_H17$ cells (**Fig 6C**). Similar results were obtained with hexanoic acid, which was also trending higher in $Cd4^{cre}Ahr^{fl/fl}$ mice (**Fig 6C**). Collectively, our data suggest that the combined effects of the cecal milieu of $Cd4^{cre}Ahr^{fl/fl}$ mice could induce changes in T cell fitness, ultimately resulting in recovery from paralysis in EAE. As such, we aimed to determine whether this held true in vivo by conducting a fecal material transfer (FMT) to C57BL/6 wild-type mice. The mice were given a cocktail of oral antibiotics for 2 weeks followed by a series of 3 oral gavages of the flushed contents of the intestinal tract from separately housed $Cd4^{cre}Ahr^{fl/fl}$ and $Ahr^{fl/fl}$ animals. During this time, the animals were also cohoused with an untreated member of the respective genotype to promote lasting engraftment of the microbiome (**Fig 6D**). When these mice underwent EAE, the wild-type mice given FMT from $Cd4^{cre}Ahr^{fl/fl}$ animals had a similar, but less pronounced reduction in disease severity as the donor mice (**Fig 6E and 6F**). Because the FMT resulted in an intermediate phenotype, we next aimed to determine whether an individual component of the gut microenvironment could act as a therapeutic for EAE. As taurocholic acid was the most efficient driver of apoptosis in vitro (**Fig 6B**), we chose this as our prime candidate. We administered taurocholic acid orally for 7 days after EAE induction (**Fig 6G**). Compared to mice given saline, those receiving taurocholic acid had decreased peak EAE scores (**Fig 6H**) and delayed onset (**Fig 6I**). These data are in support of recent work showing that bile acid receptor agonists and bile acid cocktails can reduce T cell activation and suppress EAE [36,37].

## Discussion

Here, we demonstrate that the deletion of the AHR in CD4+ cells can promote recovery in chronic EAE while having no impact on the onset nor the initial magnitude of the disease. We further show that this phenotype is dependent on the microbiome, as cohousing littermate $Cd4^{cre}Ahr^{fl/fl}$ and $Ahr^{fl/fl}$ mice abrogates recovery in EAE. These data are in support of previous work in similar models has shown no difference in EAE recovery after T cell-specific *Ahr* deletion [38]. At the mechanistic level, our data suggest that specific metabolites elevated in the cecum of $Cd4^{cre}Ahr^{fl/fl}$ mice reduce T cell fitness and viability. In particular, the cecal environment of $Cd4^{cre}Ahr^{fl/fl}$ animals is characterized by high levels of isovaleric acid and taurocholic acid, which induce T cell apoptosis in vitro, suggesting a potential mechanism for EAE recovery.

AHR activity can regulate autoimmunity via natural killer (NK) cells [39], macrophages [40], dendritic cells [29], and T cells [26,29,41]—all contributors to EAE pathogenesis. By utilizing CD4 cre mice, we will be targeting all cell types that are affected by CD4 expression at any time-including CD8 T cells that express CD4 during thymic development. AHR activation is a known regulator of T cell differentiation and inflammatory to regulatory T cell ratios. Importantly, T cell activity is dependent on the specific ligand activation with some promoting a proinflammatory environment and others having the opposite effect [26]. It has been

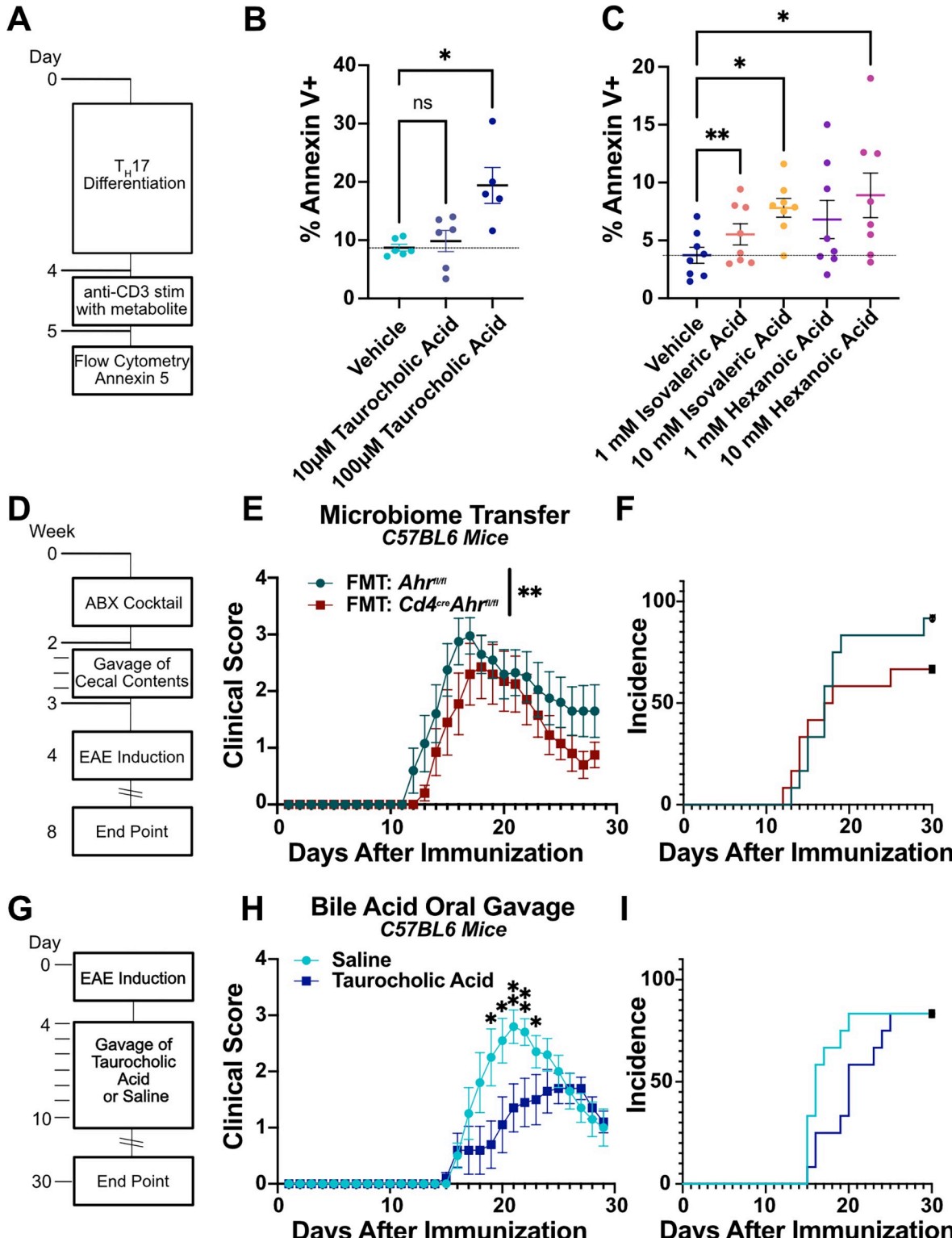

**Fig 6. Gut microenvironment and bile salts can drive EAE recovery and T cell apoptosis.** (**A**) Schematic representation of experiments in B and C. In vitro derived $T_H17$ cells from C57BL6/J mice were treated with a metabolite in addition to mild anti-CD3 stimulation for 24 h. (**B**) Taurocholic acid (10 μm and 100 μm) was administered during anti-CD3 stimulation and apoptosis of $T_H17$ cells isolated from C57BL6/J mice was measured using Annexin V staining by flow cytometry. ($n$ = 5–6 mice/group; $N$ = 2 experiments; mixed effects analysis with Geisser–Greenhouse correction [$p$ = 0.0070]; Dunnett's multiple comparison test [Veh vs. 10 μm $p$ = 0.8178, Veh vs. 100 μm

$p$ = 0.0233]). (**C**) Administration of isovaleric acid and hexanoic acid (SCFAs that are increased in mice that recover) are sufficient to drive apoptosis in $T_H17$ cells isolated from C57BL6/J mice. ($n$ = 8 mice/group; $N$ = 2 experiments; RM one-way ANOVA with the Geisser–Greenhouse correction [$p$ = 0.0399]; Dunnett's multiple comparisons test [Veh vs. 1 mM hexanoic $p$ = 0.0764, Veh vs. 10 mM hexanoic $p$ = 0.0352, Veh vs. 1 mM isovaleric $p$ = 0.0026, Veh vs. 10 mM isovaleric $p$ = 0.0254]). (**D**) Schematic depicting approach to FMT. (**E**) EAE clinical scores of C57BL6/J mice receiving FMT from separately housed $Ahr^{fl/fl}$ or $Cd4^{cre}Ahr^{fl/fl}$ mice. (Females; representative plot includes $n$ = 11 mice/group; total replicates of $N$ = 2 experiments; Mann–Whitney U test on total scores reported in legend [$p$ = 0.0096]). (**F**) Incidence of clinical sign development in mice treated with FMT. (**G**) Schematic depicting approach to oral supplement of 12.5 mg taurocholic acid/day in C57BL6/J mice immunized for EAE. (**H**) EAE clinical scores of C57BL6/J mice receiving 7 days of oral taurocholic acid (Females; $n$ = 12 mice/group; total replicates of $N$ = 1 experiment; Mann–Whitney U test on single days reported on plot). (**I**) Incidence of clinical sign development in mice treated with taurocholic acid. Error bars represent standard error from the mean. Raw data can be found in Supporting information (S1 Data). EAE, experimental autoimmune encephalomyelitis; FMT, fecal material transfer; SCFA, short-chain fatty acid.

postulated that T cell-intrinsic AHR activity is necessary for differentiation but several groups have shown that AHR antagonists do not have a cell-intrinsic effect on T cell differentiation leaving remaining questions in the field [42]. Here, we demonstrate that T cell-specific AHR expression is not necessary for the differentiation of T cells in vitro supporting the hypothesis that the AHR activity in antigen presenting cells is the necessary factor influencing T cell differentiation in vivo [27–29]. Beyond T cells, AHR activation in glial cells regulates the pathogenic effects of microglia and astrocytes in EAE via changed cytokine production [43,44]. AHR's ubiquitous expression in barrier tissues coupled with its complex ligand-binding profile warrants further study of AHR function in autoimmunity. We have identified that the changes of the luminal microenvironment of the intestine resulting from *Ahr* knockout in CD4+ cells lead to decreased post-activation lifespan in T cells; however, the impact of these changes on other cell types and other models of autoimmunity have yet to be determined. Though we aimed to target T cells using the CD4 cre line, we acknowledge that other CD4 expressing cells including some gut macrophages may also be contributing to the observed phenotype.

Bile acids, SCFAs, and other small molecules we identified can move into circulation and some can gain entry into the CNS by crossing the blood brain barrier [45]. There are likely systemic effects that have yet to be described. For example, our data suggest that significant remyelination is taking place in the chronic phase of EAE in CD4-specific *Ahr* knockout animals. Given the described impact of the microbiome and SCFA on myelination [46,47], it would be critical to test if the metabolic signature generated by the lack of *Ahr* in T cells can influence oligodendrocyte generation.

The data presented here support a bidirectional communication between T cells and the microbiome via AHR. However, it remains to be understood how T cells can modulate the metabolic signature of the cecum. We hypothesize that there are many mechanisms that could be mediating the change. For example, we show that the expression of bile acid transporters is reduced in the intestines of mice with *Ahr* deletion. This could lead to heightened levels of bile acids that have an impact on microbial viability, metabolism, and community structure. Additionally, it is possible that *Ahr* deletion in T cells may lead to modulation of IgA secretion by B cells [48,49]. IgA is the primary mechanism by which the adaptive immune system can directly modulate mucosal microbial communities [48]. An alternative hypothesis is that immune cell activity is modulating antimicrobial peptide release indirectly through communication with Paneth cells [50,51]. Most work on these systems thus far has focused on models of severe microbiome dysbiosis. Less is known about the subtler changes in the microbiome subject to homeostatic regulation. Therefore, the mechanism by which the adaptive immune system can interface with the microbiota is of great interest for understanding the basis of autoimmune disorders.

*Ahr* knockout in T cells results in a change in both primary and conjugated bile acids in the cecum. Bile acids are produced by the host for digestion and are normally reabsorbed by

transporters (primary) or passively (primary conjugated) in the ileum while secondary bile acids are not as easily absorbed [52]. There is an increase in conjugated bile acids in *Ahr* knockout mice, but lower levels of secondary bile acids. This is the inverse of patients with MS indicating that there may be a causative connection in both patients and mice with EAE [31]. Further studies are needed to understand which step of this pathway: production, conversion, or absorption are influenced by the gut flora and/or the T cells. Our results support the existing work in synthetic bile acid receptor agonists showing that T cell fitness and activity can be mediated by bile acid receptor activation [36,37]. As previous groups have shown that T cells traffic to the gut before the CNS during EAE, we hypothesize that exposure to these factors during priming in the lymph node may lead to the early apoptosis after trafficking to the CNS [53]. This evidence that bile acids can have a direct influence on T cell function warrants further study in the context of autoimmune diseases.

Microbiome-modulated immune responses become particularly important in patients with chronic and relapsing autoimmune conditions like MS [54–58]. We have described a novel function for the environmental sensor AHR in T cells as a regulator of the gut microenvironment. As we further understand the complexities of harnessing the gut microbiome, we predict tools for controlling host-derived microbe modulation will be a new frontier for therapeutics. Our work is an important step in the discovery of a targetable endogenous modulator of the microbiome and opens a potential new therapeutic avenue for MS patients.

## Materials and methods

### Experimental design

The objective of these studies was to understand the role of CD4-specific AHR activity in EAE. Immunophenotyping and characterization of the mice occurred in cohoused mice to determine the purely genetic differences. Once no major changes were discovered in the *Ahr* knockout mice, we began to separately house them based on genotype. This allowed us to examine the microbiome-dependent features of autoimmunity and the development of further hypotheses. Sample sizes were determined by power analysis. The final desired $n$ was divided by 2 and included for each experiment. In this way, we have 2 independent experiments resulting in appropriate power. Except where otherwise stated, all experiments were conducted twice independently. Data collection for all EAE experiments ended at 30 days—the time point at which we have defined as chronic stage. All experiments were conducted on age- and sex-matched animals. Histological analysis was conducted by a blinded investigator. Clinical scores are reported as the average of 2 blinded investigators to account for interrater variability. Statistical outliers identified using the ROUT method (Q = 1%) are not included in analysis or figures.

### Mice

AHR[tm3.1Bra]/J(Ahr[fx]) (#006203), B6.Cg-Tg(Cd4-cre)1Cwi/BfluJ (CD4Cre) (#022071), and C57BL/6J (#000664) mice were purchased from Jackson Laboratories. Mice were bred in-house with a 12-h light/dark schedule. Cohoused mice were housed with their littermates of the same sex from birth. Separately housed mice were separated at weaning (3 weeks) based on presence of Cre. Litters born within 4 days of each other were housed together to achieve 2 to 5 mice per cage. All procedures were approved by the University of Virginia ACUC (protocol #1918).

## Experimental autoimmune encephalomyelitis

EAE was induced in both male and female mice between 8 and 12 weeks of age. MOG$_{35-55}$ peptide (100 μg, CSBio; CS0681) was emulsified in complete Freund's adjuvant containing *Mycobacterium tuberculosis* (1 mg/mL, Sigma; F5881; M. tuberculosis (BD 231141) added to final concentration of 4 mg/mL) and was injected subcutaneously (100 μL volume) at the base of the tail. Pertussis toxin (200 ng, List Biologicals; 180) was administered i.p. on the day of and 1 day after MOG immunization. Mice were scored daily by 2 blinded evaluators using the following scale: 0-no clinical disease, 1-limp tail, 2-hindlimb incoordination, 3-hindlimb weakness, 4-hindlimb paralysis, 5-moribund. As is standard practice, animals that did not develop any signs of EAE were excluded from analysis. Total incidence including excluded animals is reported in **S1 Data**.

## Immunohistochemistry and fluorescent microscopy

Spinal cords were prepared as previously described [59]. Slides were deparaffinized using xylenes and an ethanol gradient. IHC antibodies used were anti-CD3 (Dako; A0452) at 1:200 dilution and anti-CD45 (BD Biosciences; 550539) at 1:80 dilution. Adjacent sections were stained with Luxol fast blue and hematoxylin and eosin (HE) for demyelinating plaques. In short, slides were deparaffinized using xylenes and an ethanol gradient. Slides were stained with Luxol fast blue at 56°C overnight. Slides were rinsed with distilled water and then 0.05% lithium carbonate until sharp contrast between white matter and gray matter was evident. Slides were imaged using an EVOS microscope. All images were analyzed with ImageJ by setting a threshold in the negative control and evaluating percent positive area within the white matter of the spinal cord. Luxol fast blue alone was used for quantification of myelin coverage while Luxol fast blue with HE images were used in the figure to highlight lesions. Histological analysis was performed by a blinded evaluator.

For TUNEL staining, cryosections (30 μm) were stained with anti-CD3 (eBioscience; 50-0032-82) or isotype control (Fisher Scientific; 14-0032-82) as previously described [59]. A positive control slide was treated with DNAse (3,000 U/mL). After staining, sections were fixed with paraformaldehyde for 15 min at room temperature. Sections were washed 2 times in PBS followed by proteinase K (200 μg/mL in PBS) treatment for 10 min at room temperature. Sections were washed twice with PBS and incubated with TUNEL reaction mixture (Roche; 12156792910) for 60 min at 37°C in a humidified chamber. Sections were washed 3 times in PBS and mounted using Prolong Gold Anti-Fade Reagent (Life Technologies). Slides were imaged with a Leica TCS SP8 confocal microscope. All images were analyzed with ImageJ. Histological analysis was performed by a blinded evaluator.

## RNA extraction and quantitative PCR

Whole spinal cords or 0.5 cm sections of gut tissue were homogenized using a dounce homogenizer and RNA was extracted using the Bioline Isolate II RNA mini kit as per manufacture's protocol (BIO-52073). RNA was quantified with a Biotek Epoch Microplate Spectrophotometer. Normalized RNA was reverse transcribed to cDNA with the Bioline SensiFast cDNA Synthesis Kit (BIO-65054). For the qPCR array, 2 spinal cords per group were pooled and analyzed as described by the manufacturer (Demyelinating Diseases Tier 1–4 M384; 10038953). Primers for targeted qPCR are listed in the Supporting information (S1 Data). Results were analyzed with the relative quantity (ΔΔCq) method.

## Isolation and preparation T cells

Mesenteric, inguinal, axial, and brachial lymph nodes plus the spleen were dissected from the mice after $CO_2$ euthanasia. They were then dissociated through sterile 70 μm filters and lysed with ACK lysis buffer (Quality Biological; 118-156-101). Naïve CD4+ T cells were then sorted using an EasySep Mouse CD4+ T cell Isolation Kit (Stem Cell Technologies; #19852). The cells were incubated at $1 \times 10^6$ cells/mL in skew media in anti-CD28 (InVivoMab; BE0015-5) and anti-CD3 (InVivoMab; BE0001-1) coated plates as previously described [60]. $T_H1$ and $T_H2$ media contained RPMI (Gibco; 11875–093) supplemented with 10% FBS (Optima; S12450H), 0.5% Pen/Strep (Gibco; 15240–062), 1 mM sodium pyruvate (Gibco; 11360–070), 2 mM L-glutamine (Gibco; 25030081), 10 mM HEPES (Gibco; 15603–080), 1:100 NEAA (Gibco; 11140–050), and 50 μM B-mercaptoethanol (Fisher Scientific; O3446I-100). Treg and $T_H17$ media contained IMDM (Gibco; 12440–053) using the same supplements. $T_H1$ skew media contained IL-2 (100 U/mL; Tecin, 23–6019), IL-12 (10 ng/mL; PeproTech, 210–12), and anti-IL-4 (10 μg/mL; InVivoMab, BE0045). $T_H2$ skew media contained IL-2 (100 U/mL; Tecin, 23–6019), IL-4 (10 ng/mL; InVivoMab, BE0045), and anti-IFNγ (10 μg/mL; InVivoMab BE0054). Treg skew media contained TGF-B (5 ng/mL; BioLegend; 763102) and anti-CD28 (2 μg/mL; InVivoMab; BE0015-5). $T_H17$ skew media contained IL-6 (20 ng/mL; BioLegend; 575704), IL-23 (10 ng/mL; Invitrogen; 14–8231), TGF-B (0.3 ng/mL; BioLegend; 763102), anti-IL-4 (10 μg/mL; InVivoMab; BE0045), and anti-IFNγ (10 μg/mL; InVivoMab; BE0054). $T_H1$ and $T_H2$ cell cultures were expanded on day 3 after plating by transferring cells to larger plates and adding 3× volume of RPMI supplemented media containing IL-2 (100 U/mL). On day 5, skewing was complete and cells were washed and plated for downstream assays. Treg and $T_H17$ cell cultures were not expanded but simply washed and plated for downstream assays on day 4 after plating.

## Apoptosis and cytokines production assay

Skewed T cells were plated with respective treatments at $1 \times 10^6$ cells/mL in 48-well plates coated with anti-CD3. After 24 h, the supernatant was collected for ELISA assay or cells were collected for Annexin V staining. In short, cells were stained with Ghost Live Dead (Tonbo Biosciences; 13–0870), Annexin V (Fisher Scientific; BDB556420) per manufacturer's instructions (Annexin V, fixed using the FoxP3 permeabilization, fixation kit (eBioscience; 00-5523-00)), and stained with anti-RORγt PE. Results were gated on singlets, RORγt+, and Annexin V+, L/D negative.

## Enzyme-linked immunosorbent assay

ELISA for IFNγ, IL-4, IL-17, and IL-10 were performed as previously described [61]. Antibodies used were as follows: anti-IFNγ (BioLegend; 517902), biotin-labeled anti-INFγ (BioLegend; 505704), anti-IL-4 (Thermo Fisher; 14-7041-85), biotin-labeled anti-IL-4 (Thermo Fisher; 13-7042-85), anti-IL-17 (eBioscience; 14-7175-85), biotin-labeled anti-IL-17(eBioscience; 13-7177-85), anti-IL-10 (BioLegend; 505002), and biotin-labeled anti-IL-10 (BioLegend; 505004).

## Isolation of cecal contents

For both metabolomics and in vitro analysis, the cecal contents were resuspended in 1 mL of IMDM by vortexing. Preparations were then sequentially processed using first syringe filters to remove insoluble materials (70 μm, 40 μm, 0.22 μm) and then protein concentrators (50 and 3 kDa from Thermo Fischer Scientific) to exclude macromolecules with a molecular weight >3 kDa. Preparations were used immediately or flash frozen at −80˚C for up to 3 weeks.

## Spinal cord dissociation

Spinal cords were collected in 2.5 mL ice cold HBSS (Gibco; 14025134). Equal amounts of 4 mg/mL collagenase 4 (Worthington; LS002139) with 50 U/mL DNase (Worthington; LS002139) were added and samples were shaken 3 times at 37˚C for 15 min, triturating in between. Spinal cords were strained through a 70 mm filter into 10 mL DMEM (Gibco; 11965–092) with 10% FBS (R&D Systems; S12450H). The samples were then centrifuged at 260 g for 8 min. The pellet was resuspended in a 15-mL tube in 12 mL of Percoll (GE Healthcare; 17-0891-01). The sample was then centrifuged at 650 g for 25 min without brake. The myelin layer and supernatant were aspirated and the pellet was resuspended in 2 mL of FACS buffer.

## Spinal cord luminex

Mice were perfused with PBS containing heparin (5 U/mL; Sigma; H-3125) solution. Spinal cords were dissected and flash frozen. Frozen spinal cords were homogenized in 1 mL of PBS with protease inhibitor (MedChem Express; HY-K0010) using a Dounce homogenizer. Triton X-100 was added to final concentration of 0.2% and samples were vortexed. Samples were centrifuged at 10,000 g for 5 min at 4˚C and 200 μL was acquired from the top layer to be flash frozen for later analysis. Mouse $T_H17$ Luminex was run per manufacturer's instructions on a MagPix by the University of Virginia Flow Cytometry Core, RRID: SCR_017829.

## Microbiome transfer

Female C57BL6/J mice were given drinking water containing 1 g/L Ampicillin (Sigma-Aldrich; A8351-25G), 1 g/L Neomycin (Sigma-Aldrich; N6386-25G), 1 g/L Metronidazol (Sigma-Aldrich; M1547-25G), 500 mg/mL Vancomycin (Sigma-Aldrich; V1130-5G), and 8 mg/mL Splenda for 2 weeks. Mice were then orally gavaged with fresh filtered (40 μm) cecal contents from pooled donor animals on day 0, day 2, and day 4 after removal from antibiotics. On the day of antibiotics withdrawal, a naïve microbiome donor animal was added to the cage and remained for the duration of the experiment. EAE was induced 2 weeks post reconstitution.

## Bile acid supplement

Female and male C57BL6/J mice were immunized with MOG as described above. Daily gavage of either 100 μL of 125 mg/mL taurocholic acid (in saline) or 100 μL of saline started on day 4 and continued through day 10. Mice were then monitored and scored as described earlier.

## Flow cytometry

Single-cell suspensions were incubated with Fc Block CD16/32 (Invitrogen, 14-0161-85). The cells were then incubated with a 1:200 dilution of antibody for surface stains and Live/Dead Ghost Dye Violet 510 (Tonbo Biosciences; 13–0870). Surface antibodies: 488-conjugated CD8 (Invitrogen; 53–008182), APCe780-conjugated anti-TCRβ (Invitrogen; 47-5961-82), e450-conjugated anti-CD4 (Invitrogen; 48-0042-820), 488-conjugated anti-CD11b (Invitrogen; 53-0112-82), and APC-conjugated anti-CD45.2 (Biolegend; 109813). For transcription factor staining, the eBioscience FoxP3/Transcription Factor Staining Kit (00-5523-00) was used per manufacturer's instructions with intracellular antibodies: PE-conjugated anti-RORγT (Invitrogen; 12-6981-82), PE-Cy7-conjugated anti-FoxP3 (Invitrogen; 25-5773-82), PE-conjugated anti-GATA3 (Invitrogen; 12-9966-42), PE-Cy7-conjugated anti-Tbet (Biolegend; 644824). OneComp eBeads (Thermo Fisher Scientific, 01-111-42) were used for color controls. Flow cytometry was performed using a Beckman Coulter Gallios flow cytometer and data were analyzed with FlowJo software v10.7.1.

## LC-MS

Untargeted metabolomic profiling was performed by Creative Proteomics. Cecal samples (<3 kDa) were prepared and concentrated as described before. Samples were thawed and lyophilized; 200 μL of 80% methanol was added. Samples were vortexed and then sonicated for 30 min at 4˚C. Samples were kept at −20 for 1 h, vortexed for 30 s, and kept at 4˚C for 15 min until they were centrifuged at 12,000 rpm at 4˚C for 15 min. Approximately 100 μL of the supernatant and 2.5 μL of DL-o-Chlorophenylalanine were used for LC-MS analysis. Separation is performed by Ultimate 3000 LC combined with Q Exactive MS (Thermo) and screened with ESI-MS. The LC system is comprised of ACQUITY UPLC HSS T3 (100 × 2.1 mm × 1.8 μm) with Ultimate 3000 LC. The mobile phase is composed of solvent A (0.05% formic acid water) and solvent B (acetonitrile) with a gradient elution (0 to 1.0 min, 5%B; 1.0 to 12.0 min, 5% to 95%B; 12.0 to 13.5 min, 95%B; 13.5 to 13.6 min, 95% to 5%B; 13.6 to 16.0 min, 5%B). The flow rate of the mobile phase is 0.3 mL·min-1. The column temperature is maintained at 40˚C, and the sample manager temperature is set at 4˚C. Mass spectrometry parameters in ESI+ and ESI- mode are listed as follows:

ESI+: Heater Temp 300˚C; Sheath Gas Flow rate, 45arb; Aux Gas Flow Rate, 15arb; Sweep Gas Flow Rate, 1arb; spray voltage, 3.0 KV; Capillary Temp, 350˚C; S-Lens RF Level, 30%.

ESI-: Heater Temp 300˚C, Sheath Gas Flow rate, 45arb; Aux Gas Flow Rate, 15arb; Sweep Gas Flow Rate, 1arb; spray voltage, 3.2 KV; Capillary Temp, 350˚C; S-Lens RF Level, 60%.

## LC-MS data processing

LC-MS data was processed using Rstudio. Analysis was conducted using POMA Shiny version 1.4.0 [62]. In short, we removed features with more than 20% missing values, normalized with log scaling and then completed sparse partial least squares discriminate analysis (sPLS-DA). Top ranked products were utilized in MetaMapp [63,64] analysis to produce a web of chemically and biologically related compounds. Finally, the top ranked products were also inputted into Metaboanalyst to complete pathway enrichment and determine mouse KEGG terms.

## 16S Sequencing data processing

16S Sequencing was conducted at Microbiome Insights (cohoused feces), ZymoBIOMICS (separately housed cecum), and at the University of Virginia Genomics Core (separately housed feces). Initial sequence data was analyzed using the latest version of Quantitative Insights Into Microbial Ecology 2 (Qiime2 v2021.11) [65]. Demultiplexed paired-end sequence reads were preprocessed using DADA2 [66]. The first 20 base pairs were trimmed from forward and reverse reads before they were merged to remove adaptors. Taxonomy was assigned to amplicon sequence variants (ASVs) using a naïve Bayes classifier trained on full-length 16S sequences from the latest Greengenes16S database (13_8) clustered at 99% sequence similarity. Samples were rarified before core diversity analysis. Rarefied sampling depth was 16,000 for cecal contents, 30,000 for separately housed fecal contents, and 10,411 for cohoused fecal contents. Core diversity metrics were analyzed, including number of ASVs. Nonmetric multidimensional scaling was performed in RStudio using the phyloseq package [67].

## Statistical analysis

Unless otherwise noted, all statistical tests were run on Graph Pad Prism 9 Version 9.1.0. Statistical tests and $p$ values for each comparison are listed in the figure legends or in **S1 Data**. In the figure legends, $n$ denotes the total number of animals displayed in the figure and used in

analysis. In the figure legends, $N$ denotes the total number of replicates that were completed for the given experiment. Testing level of $\alpha = 0.05$ was used throughout.

## Supporting information

**S1 Fig. Separately housed $Cd4^{cre}Ahr^{fl/fl}$ recover from EAE phase. (A)** No genotype differences in EAE incidence from the experiment shown in Fig 1A. **(B)** No differences in EAE clinical score or **(C)** incidence in cohoused male mice (males; $n = 7$–8; $N = 1$; Mann–Whitney U Test [$p = 0.5680$]). **(D)** No genotype differences in EAE incidence from the experiment shown in Fig 1D. **(E)** Separately housed male $Cd4^{cre}Ahr^{fl/fl}$ mice recover from EAE, while $Ahr^{fl/fl}$ mice maintain paralysis. (Males; $n = 7$–8/group; $N = 1$; Mann–Whitney U Test on total scores reported in legend [$p = 0.0010$] and on single days reported on plot.) **(F)** No difference in incidence of cohoused male mice. Raw data can be found in Supporting information (S1 Data). (PDF)

**S2 Fig. Lack of AHR in T cells isolated from separately housed mice does not impact differentiation capacity or cytokine production.** Naïve T cells were isolated from separately housed at weaning animals and differentiated toward $T_H1$, $T_H2$, $T_H17$, and $T_{reg}$ in vitro. Transcription factor expression was quantified by flow cytometry. **(A)** Quantification of Tbet+ cells differentiated to $T_H1$ cell type and stimulated for 24 h with either anti-CD3 (anti-CD3) or a combination of anti-CD28 and anti-CD3 (anti-CD28) ($n = 4$ mice/group; $N = 1$ experiment; Ordinary one-way ANOVA [$p = 0.2927$]). **(B)** Quantification of GATA3+ cells differentiated to $T_H2$ cell type and stimulated for 24 h with either anti-CD3 (anti-CD3) or a combination of anti-CD28 and anti-CD3 (anti-CD28) ($n = 4$ mice/group; $N = 1$ experiment; Ordinary one-way ANOVA [$p = 0.0076$] followed by Tukey's post hoc). **(C)** Quantification of RORγt+ cells differentiated to $T_H17$ cell type and stimulated for 24 h with either anti-CD3 or anti-CD28 ($n = 4$ mice/group; $N = 1$ experiment; Ordinary one-way ANOVA [$p = 0.0163$] followed by Tukey's post hoc). **(D)** Quantification of FoxP3+ cells differentiated to $T_{reg}$ cell type and stimulated for 24 h with either anti-CD3 or anti-CD28 ($n = 4$ mice/group; $N = 1$ experiment; Ordinary one-way ANOVA [$p = 0.8801$]). **(E)** IFNγ ELISA on supernatant of differentiated $T_H1$ cells stimulated for 24 h with either anti-CD3 (anti-CD3) or a combination of anti-CD28 and anti-CD3 (anti-CD28) ($n = 4$ mice/group; $N = 1$ experiment; Ordinary one-way ANOVA [$p = 0.8639$]). **(F)** IL-4 ELISA on supernatant of differentiated $T_H2$ cells stimulated for 24 h with either anti-CD3 or anti-CD28 ($n = 4$ mice/group; $N = 1$ experiment; Ordinary one-way ANOVA [$p = 0.0064$] followed by Tukey's post hoc). **(G)** IL-17a ELISA on supernatant of differentiated $T_H17$ cells stimulated for 24 h with either anti-CD3 or anti-CD28 ($n = 4$ mice/group; $N = 1$ experiment; Ordinary one-way ANOVA [$p = 0.4826$]). Raw data can be found in Supporting information (S1 Data). (PDF)

**S3 Fig. Higher apoptosis in CD4 T cells lacking AHR. (A)** Gene expression analysis (qPCR array) performed on spinal cord RNA prepared from separately housed $Cd4^{cre}Ahr^{fl/fl}$ mice and $Ahr^{fl/fl}$ littermate controls at the peak of disease (day 16) shows increased pro-apoptotic transcripts and decreased anti-apoptotic transcript (cDNA prepared from 2 spinal cords were pooled for each sample). **(B)** Gating strategy for Annexin V+ apoptotic and necrotic cells shown in show in Figs 4 and S3C and S3D. **(C)** Quantification of total dead cells in separately housed $T_H17$ differentiated cells ($n = 3$ mice/group; $N = 1$ experiment; unpaired $t$ test [$p = 0.5130$]). **(D)** T cells differentiated to $T_H1$, $T_H2$, or $T_{reg}$ in vitro from separately housed $Cd4^{cre}Ahr^{fl/fl}$ mice have higher expression of the apoptotic marker Annexin V after 24 h of CD3 stimulation. ($n = 3$/group; $N = 1$ experiment; unpaired $t$ tests [$p = 0.4276, 0.0078$,

0.0442]). Raw data can be found in Supporting information (S1 Data).
(PDF)

**S4 Fig. Only subtle microbiota population changes in *Cd4^{cre}Ahr^{fl/fl}* mice.** (**A**) PCA plot and (**B**) microbiome composition at the genus level obtained from 16S sequencing performed on feces collected from cohoused animals. (Males; $n = 6$/group) (**C**) PCA plot and (**D**) microbiome composition at the genus level obtained from 16S sequencing performed on feces collected from separately housed animals. (Males and females; $n = 10$/group) (**E**) PCA plot and (**F**) microbiome composition at the genus level obtained from 16S sequencing performed on cecum collected from separately housed animals. (Males and females; $n = 12$/group). Raw data are available in the NCBI Gene Expression Omnibus repository accession number GSE200440; https://www.ncbi.nlm.nih.gov/geo/query/acc.cgi?acc=GSE200440.
(PDF)

**S1 Data. Statistical report for all data presented in the manuscript.**
(XLSX)

**S2 Data. Raw data files from negative ion mode LC-MS corresponding to Fig 5A–5C.**
(XLSX)

**S3 Data. Raw data files from positive ion mode LC-MS corresponding to Fig 5A–5C.**
(XLSX)

# Acknowledgments

The authors wish to thank the University of Virginia Biorepository and Tissue Research facility for their support with immunohistochemistry.

The authors would also like to thank Tula Raghavan, Dr. Sanja Arandjelovic, and Dr. Sharon Walsh for critical reading and members of the University of Virginia Brain Immunology and Glia Center for scientific input.

# Author Contributions

**Conceptualization:** Andrea R. Merchak, Alban Gaultier.

**Data curation:** Andrea R. Merchak, Deniz G. Olgun, Alban Gaultier.

**Formal analysis:** Andrea R. Merchak, Deniz G. Olgun.

**Funding acquisition:** Alban Gaultier.

**Investigation:** Andrea R. Merchak, Hannah J. Cahill, Lucille C. Brown, Ryan M. Brown, Courtney Rivet-Noor, Rebecca M. Beiter, Erica R. Slogar.

**Methodology:** Andrea R. Merchak, Alban Gaultier.

**Supervision:** Alban Gaultier.

**Visualization:** Andrea R. Merchak.

**Writing – original draft:** Andrea R. Merchak, Alban Gaultier.

**Writing – review & editing:** Andrea R. Merchak, Hannah J. Cahill, Lucille C. Brown, Ryan M. Brown, Courtney Rivet-Noor, Rebecca M. Beiter, Deniz G. Olgun, Alban Gaultier.

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
