## [Editor Report · Decision Letter 0]

4 Jul 2022

Dear Dr. Gaultier, 

Thank you for submitting your manuscript entitled "T cell Aryl Hydrocarbon Receptor Activity Tunes the Gut Microenvironment to Sustain Autoimmunity and Neuroinflammation." for consideration as a Research Article by PLOS Biology.

Your manuscript has now been evaluated by the PLOS Biology editorial staff and I am writing to let you know that we would like to send your submission out for external peer review.

Once your full submission is complete, your paper will undergo a series of checks in preparation for peer review. After your manuscript has passed the checks it will be sent out for review. To provide the metadata for your submission, please Login to Editorial Manager (https://www.editorialmanager.com/pbiology) within two working days, i.e. by Jul 06 2022 11:59PM.

Kind regards,

Paula

Senior Editor

PLOS Biology

---

## [Decision Letter · Decision Letter 1]

13 Sep 2022

Dear Dr. Gaultier,

Thank you for your patience while your manuscript "T cell Aryl Hydrocarbon Receptor Activity Tunes the Gut Microenvironment to Sustain Autoimmunity and Neuroinflammation." was peer-reviewed at PLOS Biology. It has now been evaluated by the PLOS Biology editors, an Academic Editor with relevant expertise, and by several independent reviewers. 

In light of the reviews, which you will find at the end of this email, we would like to invite you to revise the work to thoroughly address the reviewers' reports.

As you will see below, the reviewers agree that the findings are interesting, but more work is needed to conclusively demonstrate microbiome-dependency. We consider that the microbiome connection should be strengthen. In particular, the reviewers suggest to address this issue by doing in vivo experiments of fecal transplant and administration of metabolites. The reviewers also have issues with the statistics and the language in the manuscript that should be solved. Please address these and the rest of the reviewers issues. 

Given the extent of revision needed, we cannot make a decision about publication until we have seen the revised manuscript and your response to the reviewers' comments. Your revised manuscript is likely to be sent for further evaluation by all or a subset of the reviewers.

**IMPORTANT - SUBMITTING YOUR REVISION**

*Re-submission Checklist*

*Published Peer Review*

*PLOS Data Policy*

*Blot and Gel Data Policy*

Sincerely,

Paula

---

Senior Editor

PLOS Biology

REVIEWS:

Reviewer #1: Microbiome and immunity.

Reviewer #2: Gut-Brain axis.

Reviewer #3: EAE/MS mouse models, microbiome/neuro. Dr. Denis Gris, Département de pharmacologie-physiologie, Université de Sherbrooke

Reviewer #1: In this manuscript, the authors test whether Ahr expression on CD4 T cells influences the clinical course of experimental autoimmune encephalomyelitis (EAE), a mouse model of multiple sclerosis (MS), and whether the microbiota plays a role in any immunomodulation. Through the use of co-housed vs separately housed Ahr sufficient mice and mice with a CD4 specific Ahr deletion (CD4ΔAhr), the authors demonstrate that loss of Ahr on CD4+ cells promotes a remission phenotype that is likely microbiota dependent. Notably, although the microbial communities change very little in CD4�Ahr mice, the cecal metabolite pool is significantly altered and some of the differentially regulated metabolites are sufficient to enhance apoptosis of CD4 Th17 cells in vitro. These data support in vivo observations that CD3+ cells are more apoptotic in the spinal column of CD4ΔAhr mice that have undergone remission. 

The observations are timely and provide new insight into pathways that could be leveraged for management of autoimmune diseases. However, there are some questions about experimental details that should be addressed, and some key experiments that could bolster the key conclusions of this paper. 

General comments:

1. One of the main claims are that the remission phenotype seen in CD4ΔAhr mice is microbiota-dependent. In order to support the first claim of microbiota-dependence, a fecal microbiota transplant (FMT) from CD4ΔAhr mice into unrelated C57BL/6 mice that transfers the remission phenotype and elevated T cell apoptosis in the CNS would be most convincing. The inclusion of a section called "Microbiome Transfer" in the Methods section suggests the authors are familiar with this technique and may have completed the experiment already. 

2. If an FMT isn't feasible, data demonstrating that the remission in cohoused Ahrfl/fl mice have a metabolite pool that resembles CD4ΔAhr mice and apoptotic cells in the CNS could be used as additional evidence that the phenotype is transmissible via the microbiome. 

3. The other main claim is that the remission phenotype is due to impaired T cell fitness downstream of altered pools of microbial-derived metabolites. Given the identification of candidate metabolites (taurocholic acid, isovaleric acid) that influence T cell viability in vitro, an in vivo experiment demonstrating the ability of one or both of these metabolites (or the filtered cecal metabolite pool, depending on experimental feasibility) to induce remission associated with increased T cell apoptosis in the CNS would be welcome. 

4. An alternative experimental approach to support the claim that "improved recovery was the result of increased T cell apoptosis after activation in the CNS" would be to use genetically engineered T cells that are resistant to apoptosis in the CD4�Ahr mice and demonstrate that these T cells are resistant to metabolite-induced apoptosis and that the remission phenotype is lost. 

Specific comments

1. Figure 2A-K: There are no differences clinical outcomes in co-housed mice, so it is unclear to this reviewer why the authors used these groups to analyze potential differences in T cell skewing and T cell numbers in MLN, PP and iLN. Are these results similar in separately housed groups?

2. Other comments related to Figure 2A-K:

- It would help readability and interpretation if the difference between co-housed vs separately housed cohorts were clearly labeled in the figure (not just in legends). Options authors could consider include the use of different color schemes, or notations ie Ahrfl/fl-CH vs CD4creAhrfl/fl-CH and Ahrfl/fl-SH vs CD4creAhrfl/fl-SH.

- Figure Legends and text describing the T cell skewing assays need to be clarified. It is currently unclear how many biological vs technical replicates were performed. (Is the data showing 6-8 individual mice with single replicates? Were all skewing assays performed on the same day or across multiple days?)

- Methods also need more details: cytokine and antibody conditions for skewing assays, time of culture, any manipulation prior to collecting supernatants for cytokine quantification, etc. This should be in addition to the existing citation to previous lab papers. 

- Can authors provide some representative flow plots for readers to evaluate gating of transcription factors?

3. Figure 2L-Q: In these panels, the authors present total cytokine expression in spinal cord homogenates taken at peak disease in separately housed mice and conclude that "these data reinforce the hypothesis that the T cell-intrinsic reduction of Ahr activity in Cd4creAhrfl/fl mice is not responsible for EAE recovery". This may be an overinterpretation given that the analysis is not CD4 intrinsic and that the data appears underpowered (n=3, 1 repeat) to detect potential differences (TNFα looks especially intriguing). At minimum, this analysis should be repeated with a larger cohort. Flow cytometric measurement of intracellular cytokine staining could be even more informative. 

4. Figure 3: Authors compare total numbers of CD4, CD8, RORγt, Foxp3, GATA3, and Tbet-expressing cells at peak and chronic phases, identifying diminished numbers of CD4+ T cells in the SC of CD4�Ahr mice at late stages (and CD11b at peak, but not chronic stages). A few extra analyses could really increase the clarity of what is happening here.

- Can authors clarify whether the RORγt, Foxp3, GATA3, and Tbet-expressing cells are pre-gated on CD4+ T cells?

- Are there any differences in the proportion of different T helper subtypes? For example, pre-gating on CD4+ T cells and then showing the frequency of Tbet vs RORγt-expressing cells could indicate whether there are defects in Th1 vs Th17 polarization. 

- Can authors include the CD11b+ quantification at both time points?

- In Figure 2, the authors suggest there are no differences in cytokine expression in the SC homogenate at peak disease. Demonstrating a difference in cytokine expression (either via SC homogenate or intracellular cytokine staining) could support the hypothesis that the activity of effector CD4+ T cells is diminished in CD4ΔAhr mice.

- Fig 3E & F both appear to have been only performed once and should be repeated if this is the case. 

- Does flow data of CD45 and CD3 staining corroborate immunohistochemistry staining shown in Fig 3B & D?

5. Figure 4: authors identify increased apoptosis in CD4+ cells of the CD4�Ahr mice. 

- Please provide information on the samples used the gene expression data in Fig 4A ie numbers of mice, were samples pooled or analysed separately, was the analysis done once or repeated? If done only once, the data would benefit from validation in a separate cohort using targeted expression analysis of genes of interest.

- Please provide representative flow plots to show viability gating from SC and in vitro assays (could be in supplement if necessary). If available, fluorescence minus one (FMO) controls would be welcome. 

- Text states that myeloid cell viability was tested, but this is not shown in Fig 4D. 

- Please provide more details on culture conditions with the cecal contents. When were the filtered <3kDa molecules added? If present from the beginning, was there any effect on Th17 polarization or proliferation, or is this a selective survival effect?

6. Figure 5: The text mentions that there were "six primary or secondary bile acids significantly increased in the cecal microenvironment of Cd4creAhrfl/fl mice". Eight are shown in the associated figure. Can the authors specify the six in the text (and perhaps provide graphs for each)? Also in the text, authors state that deoxycholic acid can be used as a proxy for changes in microbial activity - was there any difference in DCA or ratio of primary: secondary bile acids in CD4ΔAhr mice?

7. It may be worth adding a discussion of how the authors think that cecal metabolites could be linked to T cell apoptosis in the spinal cord. There is some mention that bile acids and SCFA can be found in the SC, is this the most likely explanation, or are there other possibilities worth mentioning? Is it possible circulating metabolites could gain access to the LN to influence cell viability during priming, or that circulating metabolites could impact cells migrating from the LN to the CNS? 

8. Discussion paragraph 3 (line 356-366) focuses on ways the adaptive immune system could be modulating the microbiome. The changes mentioned (IgA and AMP) might have more effect on shaping the communities, although there is little effect of the CD4-Ahr deficiency on community composition (at least at the taxonomic level shown). Could authors speculate on other mechanisms that could influence community function?

Minor comments:

* Intro line 69-70: question the authors are asking isn't very clear on first reading, consider revising (state explicitly that you're asking whether Ahr expression influences microbiome composition and/or function)

* Could a CD4cre also be acting non-specifically? In particular, populations of gut macrophages express CD4 and thus are located at the site where these metabolites are present. Some discussion of other CD4-expressing cells that could play contribute to interactions between the immune system and the microbiome may be warranted. 

* Figure 1: Can authors provide information on the number of mice the images are representative of?

* Figure legends include descriptions of the results. Legends should be re-written to focus on key experimental details. 

* It's not clear that a 2-way ANOVA is the most appropriate statistical test for quantifying the # of CD45+ or CD3+ cells in the different sections of the SC. Could authors simply sum the totals from each section and do a t-test between the genotypes?

* Error in figure legend in supplemental Figure 2E-F. Data show cecum, but the legend suggests these data are from feces.

Methods

- How were statistical outliers determined?

- Can authors clarify the statement about animals that did not develop signs of EAE being excluded from analysis - were they also excluded from the incidence curves shown in the supplement? If so, perhaps authors can transform these data into 'Day of Onset' charts as incidence should include resistant mice. 

- Microbiome transfer is listed but not performed

- For T cell skewing assays - can authors provide more details. How many mice (biological replicates), how many technical replicates? Brief description of skewing media in addition to citation. 

- When were metabolites added to the in vitro skew, and how does that impact interpretation?

- Experimental procedures mentioned in text but not described in Methods:

---Luminex 

---LFB/H&E, methods for quantifying % myelin coverage

- The text for the Immunohistochemistry and Fluorescent Microscopy section needs to be edited for clarity. 

Reviewer #2: Overall, the authors present an interesting work focusing on AHR and T cell biology in context of microbial metabolomics. However, the authors "only" provide an interesting link without any further in vivo validation, which would definitely boost the data. I would strongly suggest to validate the findings using microbial transfer studies or dietary studies supplementing bile acids or SCFA. If the authors can provide further experiments in vivo, I would recommend the paper for publishing.

Some further comments:

- The section header for the first results section is misleading. No gut microbiota composition was analyzed, therefore, the authors can not state that the gut flora is the reason for the observed differences. I strongly recommend to change the heading towards a more appropriate title.

- "Apoptosis of Ahr deficient CD4 T cell is microbiome dependent" - here as well. I would be more specific since you hypothesize that microbial metabolites are to reason. Please choose a more appropriate title.

- Please provide representative histograms or dot plots for the flow cytometry experiments within the figure.

- Line 86-90: redundant information as previously described in the introduction. Please remove.

- What was the rationale for conducting the 16s seq. in different facilities?

Reviewer #3: Comments:

The manuscript by Merchack et al. describes novel mechanisms of EAE. The authors suggest that the microbiome plays an essential role in CD4 cell balance in AHR pathway-dependent manner. Microbiome composition plays a crucial role in this context. In my opinion, these observations are novel and highly relevant to the field of autoimmunity in general and multiple sclerosis in particular. The paper is well structured and easy to read. 

Comments

I would encourage the authors to revisit the vocabulary used in the paper. Quite often, the authors use vocabulary not suited for the field of immunology. Editing by an immunologist will help to solve this issue.

Multiple comparisons need to be preceded by ANOVA or Kruskal Wallis test.

Line 60

"While most of the canonical immune sensors respond exclusively to pathogenic material….."

This is not true, as most of the immune receptors have endogenous ligands.

Line 95, 105, and thereafter

"..myeline coverage…" 

This is an odd way of putting it. Usually, people refer to "myelin content", "amount of myelin" . 

Line 110

".. active EAE scores" do the authors mean "clinical scores"?

"Representative plot 111 includes n= 9/group; N=2". What is n 9 or 2?

Although the authors correctly use non-parametric statistics comparing clinical scores, the Mann-Whitney U test is used for two groups comparisons. Adding another dimension, such as time, requires the Kruskal Wallis test.

If the authors used H&E coupled with luxol fast blue, how did they quantify the area of myelin? Usually, for quantification, luxol fast blue is done separately so that amount of stained tissue can be quantified.

Fig 1 F

For comparisons of myelin content, ANOVA will be more appropriate. Was there a difference between KO and WT at the chronic stage?

Line 127 

"T cell skewing activity" is not the correct term. Differentiation of T cells can be skewed towards different phenotypes.

 In figure 4, the authors propose that activation-induced cell death is increased in TH17 cells from KO mice. The authors need to acknowledge that anti-CD3 activation is not considered physiological. Most commonly, anti-CD28/CD3 is used. 

Line 394

In this way, we have two independent experiments resulting in appropriate power. What is statistical power in these experiments?

Line 399

"Statistical outliers are not included in analysis or figures" How were outliers determined? How many were excluded?

---

## [Editor Report · Decision Letter 2]

9 Jan 2023

Dear Dr. Gaultier,

Thank you for your patience while we considered your revised manuscript "T cell Aryl Hydrocarbon Receptor Activity Tunes the Gut Microenvironment to Sustain Autoimmunity and Neuroinflammation." for publication as a Research Article at PLOS Biology. This revised version of your manuscript has been evaluated by the PLOS Biology editors and the Academic Editor.

Based on our Academic Editor's assessment of your revision, we are likely to accept this manuscript for publication, provided you satisfactorily address the remaining points raised by the Academic Editor. The sentence in the discussion in response to Reviewer #1 to acknowledge the possibility that observations in the CD4-cre mice could reflect CD4+ macrophages should be revised to say that you can't rule out a role of AHR in other cell types developmentally affected by CD4-expression including CD8 T cells. CD4 is expressed in double positive T cells during thymic development, so AHR would be deleted in both CD4 and CD8 T cells.

Please also make sure to address the following data and other policy-related requests.

1. DATA POLICY:

Regardless of the method selected, please ensure that you provide the individual numerical values that underlie the summary data displayed in the following figure panels as they are essential for readers to assess your analysis and to reproduce it: Figures 1ACDFG, 2BCDEFGHIKLMNOPQRS, 3ABCDEFHI, 4BCDEF, 5BCDE, 6BCEFHI, and supplementary figures S1ABCDEF, S2ABCDEFG, S3ABCD, S4ABCDEF.

**Please also ensure that figure legends in your manuscript include information on where the underlying data can be found, and ensure your supplemental data file/s has a legend.**

2. Please provide a blurb which (if accepted) will be included in our weekly and monthly Electronic Table of Contents, sent out to readers of PLOS Biology, and may be used to promote your article in social media. The blurb should be about 30-40 words long and is subject to editorial changes. It should, without exaggeration, entice people to read your manuscript. It should not be redundant with the title and should not contain acronyms or abbreviations.

3. We suggest a modification in the title to make it more accessible: "The activity of the aryl hydrocarbon receptor in T cells tunes the gut microenvironment to sustain autoimmunity and neuroinflammation".

We expect to receive your revised manuscript within two weeks.

*Published Peer Review History*

*Press*

Sincerely,

Paula

---

Senior Editor,

pjaureguionieva@plos.org,

PLOS Biology

---

## [Editor Report · Decision Letter 3]

13 Jan 2023

Dear Dr. Gaultier,

Thank you for the submission of your revised Research Article "The activity of the aryl hydrocarbon receptor in T cells tunes the gut microenvironment to sustain autoimmunity and neuroinflammation" for publication in PLOS Biology. On behalf of my colleagues and the Academic Editor, Ken Cadwell, I am pleased to say that we can in principle accept your manuscript for publication, provided you address any remaining formatting and reporting issues. These will be detailed in an email you should receive within 2-3 business days from our colleagues in the journal operations team; no action is required from you until then. Please note that we will not be able to formally accept your manuscript and schedule it for publication until you have completed any requested changes.

PRESS

Sincerely,

Paula

---

Senior Editor

PLOS Biology
